# ReSplat: Degradation-Agnostic Feed-Forward Gaussian Splatting via Self-guided Residual Diffusion

**Youngho Yoon & Kuk-Jin Yoon**
Visual Intelligence Lab., KAIST, South Korea
{dudgh1732,kjyoon}@kaist.ac.kr

## Abstract

Recent advances in novel view synthesis (NVS) have predominantly focused on ideal, clear input settings, limiting their applicability in real-world environments with common degradations such as blur, low-light, haze, rain, and snow. While some approaches address NVS under specific degradation types, they are often tailored to narrow cases, lacking the generalizability needed for broader scenarios. To address this issue, we propose Restoration-based feed-forward Gaussian Splatting, named *ReSplat*, a novel framework capable of handling degraded multi-view inputs. Our model jointly estimates restored images and gaussians to represent the clear scene for NVS. We enable multi-view consistent universal image restoration by utilizing the 3d gaussians generated during the diffusion sampling process as self-guidance. This results in sharper and more reliable novel views. Notably, our framework adapts to various degradations without prior knowledge of their specific types. Extensive experiments demonstrate that ReSplat significantly outperforms existing methods across challenging conditions, including blur, low-light, haze, rain, and snow, delivering superior quality and robust NVS performance. Code is available at `https://github.com/yh-yoon/ReSplat`.

## 1 Introduction

Novel View Synthesis (NVS) is a task aimed at generating novel views of a scene from a known set of views. NVS strives to accurately estimate the geometry and appearance of a scene, enabling the rendering of realistic images from unseen perspectives. In recent years, Neural Radiance Fields (NeRF) Mildenhall et al. (2021) have revolutionized NVS by utilizing neural networks to represent scenes in a continuous volumetric format, producing highly realistic results. However, NeRF's slow rendering speed has limited its practicality, especially in real-time applications. Solutions like InstantNGP Müller et al. (2022) and TensoRF Chen et al. (2022) have addressed these speed limitations, and Gaussian Splatting Kerbl et al. (2023), introduced later, further accelerated the rendering process. By representing scenes with Gaussian ellipsoids instead of dense point samples like NeRF, Gaussian Splatting maintains competitive visual quality while enabling faster rendering.

Despite the impressive results of NeRF and Gaussian Splatting, generalizable approaches have become a major focus area. Generalizable NeRF aims to synthesize new views without retraining on each new scene, enhancing model flexibility across diverse datasets Wang et al. (2021); Yu et al. (2021b); Wang et al. (2022b); Suhail et al. (2022). Similarly, generalizable Gaussian Splatting extends this concept, offering a faster and adaptable solution for unseen scenes Charatan et al. (2023); Chen et al. (2025); Liu et al. (2025); Ziwen et al. (2024). However, these methods have primarily been developed to work on clean multi-view images captured from controlled environments.

Against this backdrop, scene reconstruction using corrupted images has gained attention. Some studies Ma et al. (2022); Wang et al. (2022a); Yoon & Yoon (2023); Wang et al. (2023); Chen et al. (2023b) are designed to handle specific types of corruption. GAURA Gupta et al. (2024), on the other hand, leverages the capacity of feed-forward NVS models to be pre-trained on large multi-view datasets, proposing a generalizable NeRF model that operates under a variety of degradations.

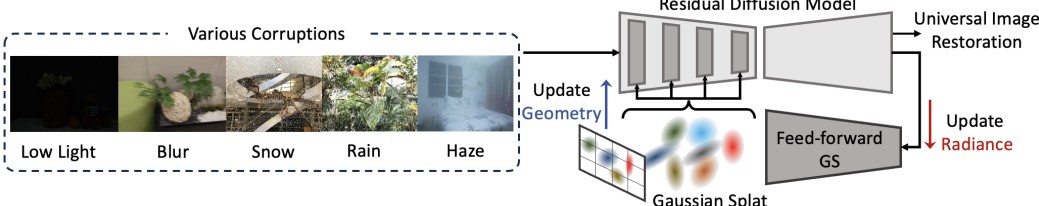

Figure 1: Proposed degradation-agnostic feed-forward Gaussian Splatting (GS) framework. Our framework achieves high-performance universal image restoration and novel view synthesis results through mutual information exchange between the universal image restoration model and the generalizable GS model.

However, GAURA excludes the image restoration capabilities developed in the 2D domain, which limits its performance potential.

Simply adopting an image restoration model does not fully address this limitation. Universal image restoration is a severely ill-posed problem, with countless possible solutions. This has led to various methods using denoising diffusion models—prominent examples of stochastic models—to address image restoration Fei et al. (2023); Özdenizci & Legenstein (2023). Research has shown that training to predict residual images enables effective image restoration Zhang et al. (2017; 2018); Zamir et al. (2021); Anwar & Barnes (2020), enhancing performance through diffusion-based residual learning.

In this paper, we propose a new generalizable gaussian splatting framework, **ReSplat**, aimed at degradation-agnostic novel view synthesis. At the core of our framework is a method that leverages the model priors of a diffusion-based unified image restoration network through Gaussian splatting. Unlike NeRF's representation, Gaussian splatting uses a point-based representation that enables explicit scene geometry extraction during training. As shown in Fig. 1, generalizable Gaussian splatting models Charatan et al. (2023); Chen et al. (2025); Liu et al. (2025) inherently estimate Gaussian centroids (geometry) using multi-view stereo (MVS) and radiance (color) through multi-view image aggregation. In our framework, a diffusion model iteratively estimates Gaussian centroids, or 3D geometry, leveraging this information to achieve 3D-consistent image restoration.

Our framework specifically adapts a 3D cross-attention module to the residual diffusion model, enabling it to utilize the location information of Gaussian centroids. Here, Gaussian centroids are derived from the point clouds of restored images estimated in the previous time-step. Second, our model performs multi-view aligned pre-filtering when generating Gaussian ellipsoids. This process involves calculating a weight map that is applied to the image features used to generate the Gaussian ellipsoids, helping to achieve artifact-free novel view synthesis. Through these techniques, our model retains the advantages of a generalizable method that operates without a scene optimization process, working effectively even in sparse-view settings while remaining degradation-agnostic. This makes it a more practical NVS model, demonstrating superior NVS and image restoration performance in multiple degradation settings compared to other approaches.

In summary, our contributions are summarized as follows:

1. We propose *ReSplat*, a novel framework for multi-view image restoration using 3DGS.
2. We introduce a multi-view aligned denoising diffusion model for universal image restoration.
3. Our method outperforms other methods in novel view synthesis and image restoration tasks.

## 2 RELATED WORKS

### 2.1 GENERALIZABLE RADIANCE FIELDS

Generating realistic images has been a central research topic for many years. Neural scene representations, such as Neural Radiance Fields (NeRF) Mildenhall et al. (2021), have emerged as effective solutions for view synthesis, achieving remarkable results. Subsequent NeRF-based approaches have further improved rendering quality Roessle et al. (2022); Wei et al. (2021); Deng et al. (2022), as well as optimization and rendering speed Sun et al. (2022); Chen et al. (2022); Fridovich-Keil et al. (2022); Yu et al. (2021a); Müller et al. (2022). However, NeRF still requires optimization for

each new scene to synthesize novel views. To address this, various studies have proposed generalizable NeRF models Yu et al. (2021b); Wang et al. (2021); Liu et al. (2022); Wang et al. (2022b); Suhail et al. (2022); Cao et al. (2022), enabling cross-scene generalization by learning a view interpolation function from source images. In these generalizable NeRFs, a common technique is to apply volume rendering for aggregating information from images, such as deep features, depth maps, or cost volumes Wang et al. (2021); Liu et al. (2022); Johari et al. (2022); Chen et al. (2021); Xu et al. (2023). GPNR Suhail et al. (2022) and GNT Wang et al. (2022b) utilize transformers to aggregate features, enhancing information interaction along a ray to directly predict RGB values for each pixel. GMT Yoon et al. (2024) enhances rendering quality by utilizing geometry-driven multi-reference texture transfer. PixelSplat Charatan et al. (2023) and MVSplat Chen et al. (2025) propose generalizable volume rendering techniques that utilize scene parameterization with 3D Gaussian primitives Kerbl et al. (2023). We note that existing generalizable radiance fields have predominantly been studied on clean images, and we aim to address this limitation by developing a universal model utilizing a residual diffusion model.

## 2.2 NOVEL VIEW SYNTHESIS WITH DEGRADATIONS

Some research has advanced novel view synthesis (NVS) by leveraging radiance fields with physics-based multi-view geometry techniques, targeting cases where train-view images require enhancement. NeRF-W Martin-Brualla et al. (2021) addresses variations in illumination and transient occlusions by relaxing strict assumptions on consistency across inputs. Deblur-NeRF Ma et al. (2022) introduces a spatially-varying blur kernel model to handle blurry inputs effectively. CROP Yoon & Yoon (2023) proposes a cross-guided optimization of radiance fields paired with multi-view image super-resolution for high-resolution NVS. RawNeRF Mildenhall et al. (2022) facilitates high-dynamic range (HDR) view synthesis by training NeRF on raw input data and generating raw-format outputs. Similarly, HDR-NeRF Huang et al. (2022) supports exposure control and HDR image synthesis by learning two distinct implicit functions: one for the radiance field and another for tone mapping. LLNeRF Wang et al. (2023) and Aleth-NeRF Cui et al. (2024) conducted research on novel view synthesis under low-light conditions. More recently, DiET-GS Lee & Lee (2025) and DiSR-NeRF Lee et al. (2024) leverage diffusion priors to improve 3D representations from degraded inputs, but they are designed for specific corruption types such as motion blur or low resolution. HQGS Lin et al. and RobustGS Wu et al. (2025) further study Gaussian Splatting under various degraded conditions and propose task-specific 3DGS pipelines to boost robustness in these scenarios. Overall, these studies do not explicitly leverage a pretrained universal image restoration model and remain tailored to specific degradation regimes or 3D configurations. In contrast, we target a degradation-agnostic framework that actively uses a pretrained universal restoration prior within a feed-forward 3DGS pipeline, so that a single model can handle diverse and mixed degradations.

## 2.3 UNIVERSAL IMAGE RESTORATION

Developing a unified model capable of handling multiple degradations has become a growing area of interest. AiRnet Li et al. (2022) introduces a module to align various distributions into a shared distribution using contrastive learning, though this approach can be challenging to train and may limit performance. IDR Zhang et al. (2023) identifies that distinct degradation types can be separated using singular value decomposition (SVD), allowing for clean image reconstruction through reformulation of singular values and vectors. PromptIR Potlapalli et al. (2024) enhances performance by employing a prompt block to capture degradation-specific features. multi-task DINO-based restoration Lin et al. (2023) and mask-based blind restoration Qin et al. (2024) exploit strong visual priors from foundation models. Adair Cui et al. (2025) and Perceive-IR Zhang et al. (2025) further improve all-in-one restoration by adaptively modeling degradation-specific frequency cues and enhancing degradation perception, respectively. Recent works have also focused on complex distortions, such as SphereSR Yoon et al. (2022) for spherical super-resolution with arbitrary projections, and adverse weather removal via spectral-based method Jeong et al. (2025). Methods such as ProRes Ma et al. (2023) and DA-CLIP Luo et al. (2023) leverage prompt learning to fully utilize the power of large-scale models. Daclip-IR Luo et al. (2024) incorporates a CLIP-based encoder to identify degradation types, extracting semantic information from distorted images to guide a diffusion model in generating high-quality outputs. DiffUIR Zheng et al. (2024) introduces selective hourglass mapping to adapt residual denoising diffusion models Liu et al. (2024) as a comprehensive image restoration approach.

## 3 METHODS

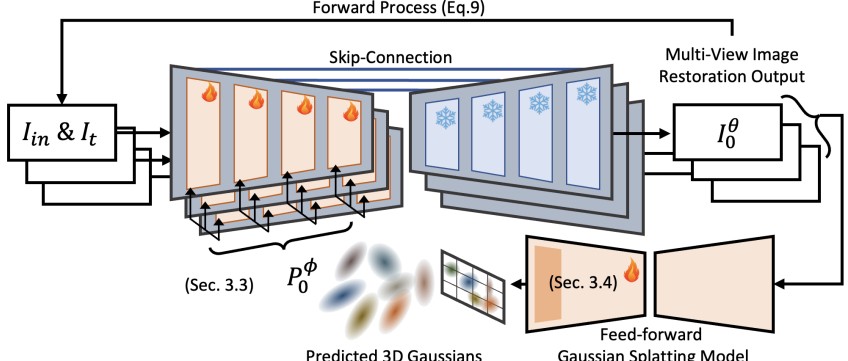

Figure 2: The overall framework for degaradation-agnostic feed-forward gaussian splatting (GS). A diffusion-based image restoration model restores the original image by iteratively estimating the residual image. During this process, feed-forward GS is performed using the original image generated in the intermediate stages of diffusion sampling. By utilizing the Gaussian points information obtained in this process, the diffusion model receives multi-view information in the next diffusion step, enabling more accurate image restoration.

In this section, we provide an overview of the residual denoising diffusion model (RDDM), universal RDDM, and 3D Gaussian splatting as preliminaries. Next, we propose an overall framework for degradation-agnostic feed-forward Gaussian splatting. Additionally, we detail two modules specifically designed to enhance NVS performance.

### 3.1 PRELIMINARIES

**3D Gaussian Splatting** 3D-GS Kerbl et al. (2023) models a scene using a collection of anisotropic 3D Gaussians, which retain the differential characteristics of volumetric representations while enabling efficient rendering through a tile-based rasterization approach. Beginning with points derived from Structure-from-Motion (SfM), each point serves as the position (mean) $\mu$ of a 3D gaussian elipsoids.

$$G(x) = e^{-\frac{1}{2}(x-\mu)^T \Sigma^{-1}(x-\mu)} \tag{1}$$

where $x$ represents a specific point in the 3D scene, and $\Sigma$ is the covariance matrix of the 3D Gaussian. $\Sigma$ is constructed from a scaling matrix $S$ and a rotation matrix $R$ with the equation $\Sigma = RSS^T R^T$. For performing tile-based rasterization, the 3D Gaussians $G(x)$ are projected onto the image plane as 2D Gaussians $G'(x)$. The rasterizer then sorts these 2D Gaussians and applies alpha blending:

$$C(x') = \sum_{i \in N} c_i \sigma_i \prod_{j=1}^{i-1}(1-\sigma_j), \quad \sigma_i = \alpha_i G'_i(x') \tag{2}$$

$x'$ represents the queried pixel position, and $N$ denotes the number of sorted 2D Gaussians associated with that pixel.

**Feed-forward 3D Gaussian Splatting** While vanilla 3DGS optimizes Gaussian parameters per scene, recent feed-forward 3DGS models Liu et al. (2025) predict them in a single forward pass from a few posed views. Given $N$ input images $\{I_{in}^i\}_{i=1}^N$ and their camera poses $\{\Pi^i\}_{i=1}^N$, the network $\phi$ maps multi-view features to per-pixel Gaussian primitives:

$$\phi : \{(I_{in}^i, \Pi^i)\}_{i=1}^N \longmapsto \{(\mu_j, \Sigma_j, \alpha_j, c_j)\}_{j=1}^{H \times W \times N}, \tag{3}$$

where $(\mu_j, \Sigma_j, \alpha_j, c_j)$ denote the center, covariance, opacity, and color of candidate Gaussians. These predictions are then pruned and merged into the explicit Gaussian set $P_\phi^0$, which is rendered using the standard splatting formulation above. In addition, the feed-forward 3DGS predicts per-view aggregation weights $W^i$ that are used to combine warped multi-view features at each novel-view pixel; in Sec. 3.4, we modulate these weights with our degradation-aware pre-filtering module.

**Algorithm 1** Training

**Input:** Clean image, Degraded image: $I_0, I_{\text{in}}$;
   GT novel view image : $I_{nv}$;
   GT residual map: $I_{\text{res}} = I_{\text{in}} - I_0$;
   Image Restoration Model: $\theta(*)$;
   Feed-forward GS Model: $\phi(*)$;

1: **repeat**
2:  $I_0 \sim q(I_0)$;
3:  $P_0^\phi = \phi(I_0, I_{in})$;
4:  $t \sim \text{Uniform}(1, \ldots, T)$;
5:  $\epsilon \sim \mathcal{N}(0, I)$;
6:  $I_t = I_0 + \bar{\alpha}_t I_{\text{res}} + \bar{\beta}_t \epsilon - \bar{\delta}_t I_{\text{in}}$;
7:  Take the gradient descent step on

$$\nabla_\theta \| I_{\text{res}} - I_{\text{res}}^\theta(P_0^\phi, I_t, I_{\text{in}}, t)\|_1 +$$
$$\nabla_\phi \| I_{nv} - I_{\text{ren}}^\phi(I_{\text{in}} - I_{\text{res}}^\theta, I_{\text{in}})\|_1;$$

8: **until** converged

**Algorithm 2** Sampling

**Require:** Degraded image: $I_{in}$;
   Image Restoration Model: $\theta(*)$;
   Feed-Forward GS Model: $\phi(*)$;

1: $\epsilon \sim \mathcal{N}(0, I)$;
2: $I_T = (1 - \bar{\delta}_T)I_{in} + \bar{\beta}_T \epsilon$;
3: $P_0^\phi = None$;
4: **for** $t = T, \ldots, 1$ **do**
5:  **if** $t > 1$ **then**
6:   $I_{t-1} = I_t - \alpha_t I_{res}^\theta(P_0^\phi, I_t, I_{in}, t)$
     $+ \delta_t I_{in}$;
7:   $P_0^\phi = \phi(I_{\text{in}} - I_{\text{res}}^\theta, I_{in})$;
8:  **else**
9:   $I_{t-1} = I_{in} - I_{res}^\theta(P_0^\phi, I_t, I_{in}, t)$;
10:   $I_{nv} = I_{\text{ren}}^\phi(I_{\text{in}} - I_{\text{res}}^\theta, I_{in})$;
11: **return** $I_0, I_{nv}$

**Residual Denoising Diffusion Model** RDDM Liu et al. (2024) uses a standard $T$-step diffusion model that includes both a forward and a reverse process. In the forward process, one-step noising is formulated as a Markov chain:

$$q(I_t|I_{t-1}, I_{res}) = \mathcal{N}(I_t; I_{t-1} + \alpha_t I_{res}, \beta_t^2 \mathbf{I}) \tag{4}$$

where $\alpha_t$ and $\beta_t$ are the noise coefficients for $I_{\text{res}}$ and gaussian noise. $I_t$ is the result at timestep $t$, and $I_{\text{res}}$ represents the residual between the degraded image $I_{\text{in}}$ and the clean image $I_0$, with $I_{\text{res}} = I_{\text{in}} - I_0$. In the reverse process, RDDM approximates the true generative distribution $p_\theta(I_{t-1}|I_t)$ by using $q(I_{t-1}|I_t, I_0, I_{\text{res}})$, which is also formulated as a Markov chain when deterministic implicit sampling using DDIM Song et al. (2020):

$$p_\theta(I_{t-1}|I_t) = \mathcal{N}\left(I_{t-1}; I_0^\theta + \overline{\alpha}_{t-1} I_{res}^\theta + \overline{\beta}_{t-1} \epsilon^\theta, 0 \cdot \mathbf{I}\right) \tag{5}$$

In summary, the relatios between $I_t$ and $I_{t-1}$ in both the forward and reverse processes is as follows:

$$I_t = I_{t-1} + \alpha_t I_{res} + \beta_t \epsilon_{t-1} \tag{6}$$

$$I_{t-1} = I_t - (\bar{\alpha}_t - \bar{\alpha}_{t-1})I_{res}^\theta - (\bar{\beta}_t - \bar{\beta}_{t-1})\epsilon^\theta \tag{7}$$

**Universal Residual Denoising Diffusion Model** DiffUIR Zheng et al. (2024) utilizes the conditioning mechanism from RDDM and incorporate a shared distribution term (SDT) within the diffusion algorithm for universal image restoration. They adjust the forward process as follows:

$$I_t = I_{t-1} + \alpha_t I_{res} + \beta_t \epsilon_{t-1} - \delta_t I_{in} \tag{8}$$

where $\delta_t I_{\text{in}}$ represents the SDT, and $\delta$ is the shared distribution coefficient. The reverse process is also as follows:

$$I_{t-1} = I_t - \alpha_t I_{res}^\theta + \delta_t I_{\text{in}} - (\beta_t^2/\bar{\beta}_t)\epsilon^\theta \tag{9}$$

Finally, in the deterministic implicit sampling process, $I_{t-1}$ and the pseudo clean image $I_0^\theta$ can be derived using the following equation:

$$I_{t-1} = I_0^\theta + \overline{\alpha}_{t-1} I_{res}^\theta - \overline{\delta}_{t-1} I_{in} \quad \text{s.t.} \quad I_0^\theta = I_{in} - I_{res}^\theta \tag{10}$$

## 3.2 OVERALL FRAMEWORK

We aim to develop a novel view synthesis model that can be performed under arbitrary degradation. Most of the existing novel view synthesis studies have been conducted on clean images without corruption, and even in the case of studies on situations with corruption, models specialized for specific degradation types are being developed Ma et al. (2022); Wang et al. (2022a); Yoon & Yoon (2023); Wang et al. (2023); Chen et al. (2023b). These studies solve the problem by simultaneously optimizing scene optimization and physical characteristics by implementing physical characteristics that

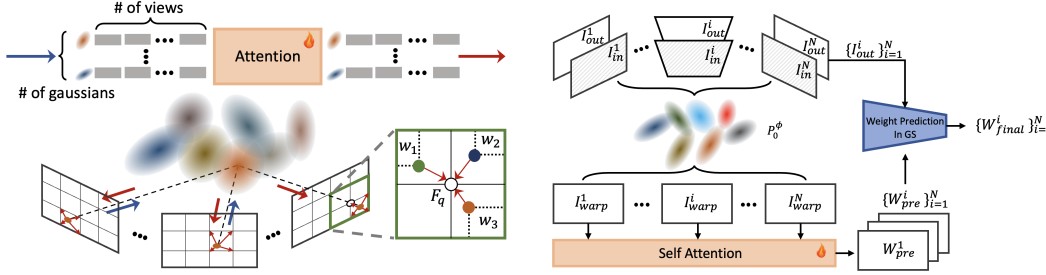

Figure 3: **GS-guided multi-view alignment**. Module embedded in the residual diffusion model that shares info between adjacent views using Gaussian centers.

Figure 4: **Pre-filtering with warped features**. Warped inputs are self-attended to form pre-filtering weights for feature aggregation.

cause specific degradation as a rendering process. Therefore, there is a need for new degradation-agnostic novel view synthesis (NVS) studies. To address this, we propose a new framework, *ReSplat*, the NVS model that leverages the diffusion prior studied in the field of 2D image restoration.

**Training stage** As shown in Fig. 2, we combine the feed-forward gaussian splatting (GS) model and the universal image restoration (UIR) model. The GS and UIR models support complementary roles. Unlike NeRF, feed-forward GS inevitably performs multi-view stereo (MVS) within the model because it needs to explicitly extract point clouds. This enables acquisition of 3D scene geometry information and helps the UIR model find corresponding points for adjacent multi-view images. Meanwhile, the UIR model performs degradation-agnostic image restoration to help feed-forward GS perform NVS using images with corruption removed. In addition, we adopt DiffUIR Zheng et al. (2024), a diffusion model-based UIR model, to perform iterative scene geometry extraction and iterative image refinement so that the UIR results can be gradually refined. The training process can be found in Algorithm 1. The first term $\left\| I_{\text{res}} - I_{\text{res}}^{\theta}(P_0^{\phi}, I_t, I_{\text{in}}, t) \right\|_1$ corresponds to the universal restoration loss $\mathcal{L}_{\text{UIR}}$, which supervises the residual prediction of the UIR model $\theta$. The second term $\left\| I_{nv} - I_{\text{ren}}^{\phi}(I_{\text{in}} - I_{\text{res}}^{\theta}, I_{\text{in}}) \right\|_1$ defines the novel-view rendering loss $\mathcal{L}_{\text{NV}}$, which supervises the feed-forward GS model $\phi$ using the ground-truth clean novel view $I_{nv}$.

**Sampling stage** The specific sampling process of ReSplat is shown in Algorithm 2. Given $N$ multi-view input images $\{I_{in}^i\}_{i=1}^N$, the $t^{th}$ noise images $\{I_t^i\}_{i=1}^N$ are generated according to the forward process of the DiffUIR. We generate predicted clean images $\{I_0^i\}_{i=1}^N$ from predicted $\{I_{res}^i\}_{i=1}^N$ by the UIR model. The predicted clean images are used to generate explicit point cloud $P_0^{\phi}$ by the MVS module of the feed-forward GS model. Meanwhile, we generate $\{I_{t-1}^i\}_{i=1}^N$ to perform the next diffusion step. After that, we perform a 3d aligned diffusion reverse process using $P_0^{\phi}$ (sec. 3.3). Through this, we regenerate the refined $\{I_0^i\}_{i=1}^N$ and $P_0^{\phi}$. We repeat the process and perform the feed-forward GS overall process using the finally generated $\{I_0^i\}_{i=1}^N$. In this process, we perform a feature pre-filtering process conditioned on the original corrupted images $\{I_{in}^i\}_{i=1}^N$ to remove points where artifacts exist before the multi-view feature aggregation process, thereby generating a more robust GS output (sec. 3.4). Through this, we can obtain a rendered output for the novel view point.

### 3.3 GS GUIDED MULTI-VIEW ALIGNMENT

Since the original UIR model is designed for a single image, it is necessary to design a module for enabling multi-view image interaction. As shown in Fig. 3, we adapt a module that performs feature attention in space to UIR by utilizing $P_0^{\phi}$, a pseudo geometry generated during the sampling process. Multi-view features are projected toward each gaussian center in $P_0^{\phi}$. When there are $N$ multi-view feature vectors $\{f_i^j\}_{j=1}^N$ for the $i^{th}$ center point $p_i$, we perform self-attention between the corresponding vectors. This process is repeated in the encoder of the diffusion model and helps ensure the 3D consistency of multi-view images. The processed feature vector $f_{i,rep}^j$ is reprojected to the original pixel coordinates. However, since the reprojected point is located in continuous coordinates, not discrete coordinates, it is necessary to propagate to the surrounding discrete coordinates. Therefore, we perform a weighted sum by applying 2D interpolation weights $\{w_i\}$ to all reprojected points existing between adjacent pixels. Each weight is determined by the area of the opposite re-

Table 1: Novel View Synthesis (NV) and Image Restoration (IR) results of five corruption types on LLFF degradation dataset with **three multi-view** inputs. The best scores and second best scores are highlighted with their respective colors.

| Method | Operation | Year | Corruption Type | Novel View Synthesis | | | Multi-View Image Restoration | | |
|---|---|---|---|---|---|---|---|---|---|
| | | | | PSNR(↑) | SSIM(↑) | LPIPS(↓) | PSNR(↑) | SSIM(↑) | LPIPS(↓) |
| AiRnet | IR → NV | 2022 | Motion Blur | 20.11 | 0.6896 | 0.4250 | 21.99 | 0.7543 | 0.3769 |
| PromptIR | IR → NV | 2023 | | 20.04 | 0.6872 | 0.4208 | 22.14 | 0.7526 | 0.3668 |
| GAURA | Only NV | 2024 | | 21.28 | 0.7198 | 0.4343 | - | - | - |
| DiffUIR | IR → NV | 2024 | | 22.75 | 0.7824 | 0.3209 | 26.34 | 0.8640 | 0.2951 |
| ReSplat | IR w/ NV | 2025 | | 23.15 | 0.8049 | 0.3151 | 27.14 | 0.8850 | 0.2713 |
| AiRnet | IR → NV | 2022 | Snow | 20.22 | 0.6852 | 0.3026 | 21.57 | 0.8184 | 0.2159 |
| PromptIR | IR → NV | 2023 | | 20.54 | 0.7067 | 0.2737 | 23.21 | 0.8578 | 0.1912 |
| GAURA | Only NV | 2024 | | 20.48 | 0.7044 | 0.3195 | - | - | - |
| DiffUIR | IR → NV | 2024 | | 24.24 | 0.8549 | 0.1826 | 31.20 | 0.9627 | 0.1019 |
| ReSplat | IR w/ NV | 2025 | | 24.46 | 0.8614 | 0.1677 | 32.07 | 0.9685 | 0.0886 |
| AiRnet | IR → NV | 2022 | Haze | 9.159 | 0.3841 | 0.3892 | 8.871 | 0.4155 | 0.2949 |
| PromptIR | IR → NV | 2023 | | 9.784 | 0.4651 | 0.3508 | 9.585 | 0.5251 | 0.2280 |
| GAURA | Only NV | 2024 | | 17.22 | 0.7205 | 0.4516 | - | - | - |
| DiffUIR | IR → NV | 2024 | | 21.56 | 0.8392 | 0.1857 | 25.57 | 0.9612 | 0.0701 |
| ReSplat | IR w/ NV | 2025 | | 21.99 | 0.8471 | 0.1750 | 26.45 | 0.9680 | 0.0619 |
| AiRnet | IR → NV | 2022 | Low-light | 9.526 | 0.1364 | 0.6041 | 6.388 | 0.0859 | 0.7654 |
| PromptIR | IR → NV | 2023 | | 6.367 | 0.0805 | 0.6240 | 6.298 | 0.0805 | 0.5789 |
| GAURA | Only NV | 2024 | | 15.28 | 0.6627 | 0.5177 | - | - | - |
| DiffUIR | IR → NV | 2024 | | 18.87 | 0.8241 | 0.2429 | 21.88 | 0.9374 | 0.1647 |
| ReSplat | IR w/ NV | 2025 | | 19.76 | 0.8276 | 0.2433 | 22.82 | 0.9452 | 0.1605 |
| AiRnet | IR → NV | 2022 | Rain | 20.49 | 0.6988 | 0.3416 | 23.09 | 0.8008 | 0.3018 |
| PromptIR | IR → NV | 2023 | | 20.71 | 0.7175 | 0.2992 | 24.78 | 0.8555 | 0.2499 |
| GAURA | Only NV | 2024 | | 21.78 | 0.7578 | 0.4110 | - | - | - |
| DiffUIR | IR → NV | 2024 | | 23.51 | 0.8313 | 0.2538 | 29.69 | 0.9357 | 0.1919 |
| ReSplat | IR w/ NV | 2025 | | 24.11 | 0.8505 | 0.2140 | 31.28 | 0.9538 | 0.1533 |

gion, ensuring that features closer to the query point have a higher influence. Therefore, when there is a discrete point $q$, the multi-view feature $F_q$ that $q$ obtains is as follows.

$$F_q = \sum_i w_i f_{i,rep}^j \quad where \quad i \in Q \tag{11}$$

and $Q$ is the set of the index of all points that exist within the smallest rectangle surrounding the point $q$.

## 3.4 PRE-FILTERING WITH WARPED FEATURES

The final outputs of the UIR, $\{I_{out}^i\}_{i=1}^N$, are first depth-warped toward the novel pose using $P_0^\phi$. The feed-forward GS backbone then produces per-view aggregation weights $\{W_{final}^i\}_{i=1}^N$ for combining the $N$ warped multi-view features at each novel-view pixel. Since these weights have a critical impact on determining the radiance of the final Gaussian ellipsoids, we introduce a pre-filtering module that is additionally conditioned on the corrupted inputs $\{I_{in}^i\}_{i=1}^N$.

As illustrated in Fig. 4, the pre-filtering module takes the warped restored and degraded images as input and predicts a per-view reliability map $\{W_{pre}^i\}_{i=1}^N$, independently of the occlusion-based weights from the GS model. We then modulate the original GS weights by this reliability map:

$$W_{final}^i(x) = W_{pre}^i(x) \cdot W^i(x), \tag{12}$$

and use the updated $W_{final}^i$ in the splatting renderer. In other words, the pre-filtering module acts as a soft, degradation-aware gate on top of the standard visibility weights: regions where residual artifacts (e.g., remaining rain streaks, snow blobs, or haze fragments) are strong or inconsistent across views receive lower $W_{pre}^i$ and are down-weighted, while geometry-consistent, clean structures are preserved, leading to a more robust radiance field and improved NVS quality. In practice, we simply replace the original per-view aggregation weights $W^i$ with the updated $W_{final}^i$. These weights are then used in its standard multi-view feature aggregation and Gaussian rendering pipeline to determine the contribution of each input view at every novel-view location.

## 4 EXPERIMENTS

### 4.1 EXPERIMENT SETTINGS

**Datasets** For training our model, we utilize the synthetic multi-degradation generation pipeline proposed by GAURA Gupta et al. (2024) to construct a multi-view degradation dataset. We use a training dataset provided by IBRNet Wang et al. (2021), commonly used in novel view synthesis task.

Table 2: Novel View Synthesis results and multi-view image restoration results of three types (rain+motion blur, snow+motion blur, and haze+snow) on LLFF **mixed degradation** dataset with three multi-view inputs. The best scores are highlighted.

| Method | Corruption Type | Novel View Synthesis | | | Multi-View Image Restoration | | |
|---|---|---|---|---|---|---|---|
| | | PSNR(↑) | SSIM(↑) | LPIPS(↓) | PSNR(↑) | SSIM(↑) | LPIPS(↓) |
| DiffUIR | Rain+Motion Blur | 20.07 | 0.6910 | 0.4885 | 20.41 | 0.7083 | 0.4653 |
| ReSplat | | 20.44 | 0.7090 | 0.4555 | 20.74 | 0.7220 | 0.4507 |
| DiffUIR | Snow+Motion Blur | 21.63 | 0.7407 | 0.4076 | 22.51 | 0.7757 | 0.3848 |
| ReSplat | | 22.00 | 0.7594 | 0.3782 | 22.90 | 0.7908 | 0.3661 |
| DiffUIR | Haze+Snow | 15.38 | 0.6978 | 0.3488 | 15.52 | 0.7702 | 0.2843 |
| ReSplat | | 20.17 | 0.7730 | 0.3148 | 19.92 | 0.8067 | 0.2808 |

Figure 5: Comparsions of novel view synthesis results of five types (motion blur, snow, haze, low-light, rain) on LLFF degradation dataset.

The test sets are divided into synthetic and real-world datasets. The synthetic dataset is generated using synthetic degradations applied to the LLFF Mildenhall et al. (2019) dataset. For real-world scenarios, we evaluate our model using the DeblurNeRF Ma et al. (2022) dataset for motion blur, the REVIDE Zhang et al. (2021) dataset for haze, and the LLNeRF Wang et al. (2023) dataset for low-light conditions.

**Network** We use DiffUIR Zheng et al. (2024), a residual diffusion model, as our baseline for image restoration. We also use MVSGaussian Liu et al. (2025), one of the state-of-the-art models, as the feed-forward GS. To accelerate the training process, MVSGaussian is first trained on our training dataset without image restoration process.

We conduct a comparison of universal image restoration with AiRnet, PromptIR, and DiffUIR. The goal of our approach is to develop an adapter that transforms a UiR model to handle multi-view inputs. Therefore, we utilize a network pretrained with a single-view UiR. For a fair comparison, all models are fine-tuned on our training dataset.

During the inference time, ReSplat uses DDIM sampling strategy with a total of three sampling steps fixed. The inference process for the three multi-view inputs can be completed within one second. For more details, please refer to the supplementary material.

Table 3: Novel view synthesis results of three types (motion blur, haze, low-light) on real-world degradation datasets.

| Method | Type | Novel View Synthesis | | |
|---|---|---|---|---|
| | | PSNR(↑) | SSIM(↑) | LPIPS(↓) |
| AiRnet | Motion Blur | 18.59 | 0.6429 | 0.4009 |
| PromptIR | | 18.42 | 0.6289 | 0.3959 |
| GAURA | | 21.54 | 0.7711 | 0.3909 |
| DiffUIR | | 22.76 | 0.8090 | 0.2988 |
| ReSplat | | 22.91 | 0.8145 | 0.2922 |
| AiRnet | Haze | 15.91 | 0.7189 | 0.3290 |
| PromptIR | | 15.32 | 0.7106 | 0.3224 |
| GAURA | | 16.90 | 0.8397 | 0.3920 |
| DiffUIR | | 17.26 | 0.8451 | 0.1900 |
| ReSplat | | 17.75 | 0.8511 | 0.1968 |
| AiRnet | Low-light | 9.526 | 0.1364 | 0.6040 |
| PromptIR | | 17.10 | 0.8322 | 0.5091 |
| GAURA | | 19.07 | 0.8503 | 0.6301 |
| DiffUIR | | 22.00 | 0.8165 | 0.4958 |
| ReSplat | | 22.92 | 0.8578 | 0.4759 |

Table 4: Ablation study with Novel View Synthesis results of 5 types (motion blur, snow, haze, low-light, rain) on synthetic degradations. Values represent the average of five degradations. The best scores and second best scores are highlighted.

| Model # | Alignment | Pre-Filtering | Novel View Synthesis | | |
|---|---|---|---|---|---|
| | | | PSNR(↑) | SSIM(↑) | LPIPS(↓) |
| 1 | X | X | 22.19 | 0.8264 | 0.2372 |
| 2 | X | O | 22.35 | 0.8290 | 0.2368 |
| 3 | O | X | 22.46 | 0.8313 | 0.2306 |
| 4 | O | O | 22.69 | 0.8383 | 0.2230 |

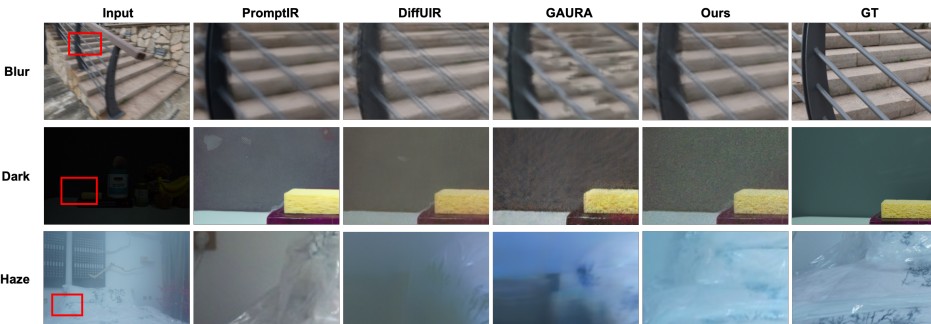

Figure 6: Visual Comparsions of novel view synthesis results of 3 types (motion blur, haze, low-light) on real-world degradation dataset (DeblurNeRF, REVIDE, and LLNeRF dataset).

## 4.2 QUANTITATIVE ANALYSIS

**Synthetic Degradation** As shown in Table 1, we evaluate ReSplat against baselines (AiRnet, PromptIR, DiffUIR, GAURA) across five corruption types: motion blur, snow, haze, low-light, and rain, for both novel view synthesis and multi-view image restoration. Performance is measured using PSNR, SSIM, and LPIPS. For novel view synthesis, ReSplat consistently outperforms other methods, especially in motion blur, snow, and rain scenarios, producing sharper, more perceptually accurate views. In multi-view image restoration, ReSplat excels in high-corruption cases, achieving the best overall visual fidelity and structural similarity. Notably, it handles complex degradations like heavy rain and motion blur more effectively than competing models, preserving both fine details and global consistency.

**Mixed Degradation** Table 2 presents a comparison between our method, ReSplat, and the strongest baseline, DiffUIR, under various mixed degradation scenarios, including Rain+Motion Blur, Snow+Motion Blur, and Haze+Snow. ReSplat consistently achieves the best performance across all conditions, significantly outperforming DiffUIR in both novel view synthesis and multi-view image restoration. We conduct a direct comparison between the top-performing method (ReSplat) and the next best (DiffUIR) to highlight the effectiveness of our approach. Notably, in more challenging scenarios such as Snow+Motion Blur and Haze+Snow, ReSplat delivers considerably higher fidelity, as reflected by higher PSNR and SSIM values and lower LPIPS.

**Real-World Degradation** As shown in Table 3, we evaluate ReSplat against baselines for novel view synthesis under real-world corruptions: motion blur (DeblurNeRF Ma et al. (2022)), haze (REVIDE Zhang et al. (2021)), and low-light (LLNeRF Wang et al. (2023)) dataset. ReSplat achieves the best overall results for motion blur, preserving structural and perceptual quality. In haze, it yields the lowest LPIPS, indicating superior perceptual quality despite similar PSNR/SSIM scores with DiffUIR. Under low-light conditions, ReSplat balances structural integrity and perceptual fidelity, performing competitively across all metrics. In addition, Table 5 and Table 6 report results on in-the-wild rain (NTURain Chen et al. (2018)) and snow (RSVD Chen et al. (2023a)) datasets, where ReSplat consistently outperforms UIR and GS baselines, demonstrating robust generalization.

Table 5: Novel View Synthesis (NV) results of **Rain** corruption on the real-world deraining dataset with three multi-view inputs. The best scores and second best scores are highlighted.

| Method | Type | PSNR(↑) | SSIM(↑) | LPIPS(↓) |
|--------|------|---------|---------|----------|
| AiRnet | | 24.05 | 0.8183 | 0.1955 |
| PromptIR | | 24.23 | 0.8230 | 0.1801 |
| GAURA | Rain | 19.39 | 0.6602 | 0.3987 |
| DiffUIR | | 23.99 | 0.8145 | 0.2094 |
| ReSplat | | 24.35 | 0.8232 | 0.1772 |

Table 6: Novel View Synthesis (NV) results of **Snow** corruption on the real-world desnowing dataset with three multi-view inputs. The best scores and second best scores are highlighted.

| Method | Type | PSNR(↑) | SSIM(↑) | LPIPS(↓) |
|--------|------|---------|---------|----------|
| AiRnet | | 20.23 | 0.7035 | 0.3103 |
| PromptIR | | 21.27 | 0.7192 | 0.3007 |
| GAURA | Snow | 20.22 | 0.7578 | 0.3647 |
| DiffUIR | | 22.12 | 0.8215 | 0.2277 |
| ReSplat | | 22.45 | 0.8263 | 0.2175 |

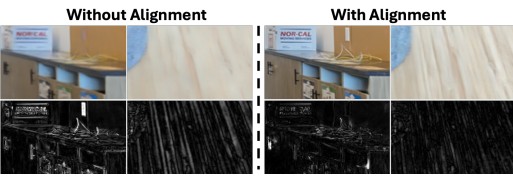

Figure 7: Qualitative comparison of the alignment module. The top row shows the restored RGB outputs, while the bottom row visualizes the corresponding error maps.

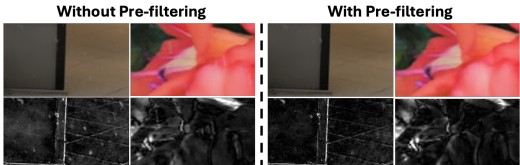

Figure 8: Qualitative comparison of pre-filtering module. The top row shows the restored RGB outputs, while the bottom row visualizes the corresponding error maps.

### 4.3 Qualitative Comparison and Analysis

**Synthetic Degradation** As shown in Fig. 5, our method surpasses existing baselines across all tested degradations. It achieves higher visual fidelity and color accuracy, especially in challenging haze and low-light scenarios, closely matching the ground truth.

**Real-World Degradation** As shown in Fig. 6, our method robustly handles real-world degradations, producing sharper results with superior color fidelity. Unlike competing approaches that introduce noise, ours effectively reconstructs fine structures in challenging scenarios.

### 4.4 Ablation Studies

We conduct an ablation study using four model variants: Model 1 as baseline, Model 2 with prefiltering, Model 3 with alignment, and Model 4 with both. Metrics are averaged over five LLFF degradation datasets. As shown in Table 4, each component contributes to improved performance in novel view synthesis. Adding pre-filtering (Model 2) increases PSNR and reduces LPIPS, indicating a modest improvement in reconstruction quality. Alignment alone (Model 3) further enhances PSNR and other quality metrics compared to the baseline. When both alignment and pre-filtering are applied (Model 4), the model achieves the best overall results, with a PSNR of 22.69, demonstrating a clear effect across various degradations. Qualitative comparisons in Fig. 7 and Fig. 8 illustrate that alignment reduces multi-view geometric inconsistencies, while pre-filtering suppresses residual artifacts without destroying fine structures in the rendered novel views.

## 5 Limitations

Despite its performance, ReSplat has several limitations. The diffusion refinement adds computational and memory overhead, requiring further optimization for high-resolution scaling. Additionally, it inherits 3DGS's biases in handling specularities and transparency. While dependent on a pretrained restoration prior, this modularity allows ReSplat to benefit from future restoration advancements without architectural changes.

## 6 Conclusion

We present a feed-forward gaussian splatting framework for degradation-agnostic novel view synthesis. By integrating a residual diffusion model with 3D cross-attention and multi-view prefiltering, our method robustly restores images and improves geometry estimation, outperforming existing approaches in both novel view synthesis and universal multi-view image restoration.

ACKNOWLEDGEMENTS

This work was supported by the Technology Innovation Program (2410013617,20024355, Development of autonomous driving connectivity technology based on sensor-infrastructure cooperation) funded By the Ministry of Trade, Industry & Energy(MOTIE, Korea) and the Institute of Information & communications Technology Planning & Evaluation (IITP) grant funded by the Korea government(MSIT) (No. RS-2024-00457882, AI Research Hub Project).

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
