# OpenReview forum: "ReSplat: Degradation-agnostic Feed-forward Gaussian Splatting via Self-guided Residual Diffusion"
_ICLR.cc/2026/Conference — ICLR 2026 Poster_

### Official Review · Reviewer_26dS · 2025-10-16

**Soundness:** 3
**Presentation:** 2
**Contribution:** 3
**Rating:** 6
**Confidence:** 4

**Summary:**

1. **Originality-wise**: The core idea of creating a synergistic loop between a universal image restoration model and a feed-forward Gaussian Splatting model is highly novel. Specifically, using the intermediate 3D geometry from the GS model to enforce multi-view consistency in a diffusion-based restoration process is a clever and previously unexplored mechanism for this problem.
2. **Quality-wise**: The claims are strongly supported by comprehensive experiments across a wide array of synthetic, mixed, and real-world degradations. The method consistently achieves state-of-the-art results in both novel view synthesis and image restoration tasks, demonstrating the framework's robustness and effectiveness.
3. **Clarity-wise**: The paper is well-structured and clearly articulates a complex problem and its solution. The overall framework is well-illustrated with diagrams that effectively convey the interplay between the restoration and synthesis modules. The motivation, methodology, and results are presented logically and are easy to follow.

**Strengths:**

1. **Solves a Novel and Practical Problem**: The work addresses degradation-agnostic novel view synthesis, a topic of high practical importance for real-world applications that has been underexplored compared to synthesis from clean images.

2. **Innovative Synergistic Framework**: The core contribution is a novel framework where an image restoration model and a Gaussian Splatting model work in tandem. The use of 3D geometry from the GS model to guide the UIR model and ensure multi-view consistency is a key innovation.

3. **SOTA Performance and Thorough Validation**: The method demonstrates superior performance on both novel view synthesis and image restoration tasks across a comprehensive set of experiments, including single, mixed, and real-world degradations.

**Weaknesses:**

1. Ambiguous Mechanism and Potential for Detail Suppression in the Pre-filtering Module: The paper proposes a pre-filtering module to suppress artifacts but provides insufficient insight into its inner workings. The mechanism, which uses self-attention on both the corrupted and restored images, is a black box. It is unclear whether the module learns to identify specific "restoration artifacts" or if it simply learns to penalize any high-frequency regions that differ significantly from the degraded input.

2. the complete absence of failure cases is a significant omission for a paper claiming such robust, "agnostic" capabilities. A rigorous scientific contribution requires a transparent discussion of a method's limitations.
---
I have listed my concerns, and the score will be adjusted based on the author's response.

**Questions:**

Please refer to Weaknesses part.

---

> ### Author Response · Authors · 2025-11-24
>
> W1. Thank you for raising this concern. The pre-filtering module takes warped restored features together with the corresponding degraded features and produces a per-view reliability map that *modulates* the original GS aggregation weights, so it acts as a soft gate on top of the standard visibility weights. By using both degraded and restored inputs across multiple views, it is biased to down-weight view-inconsistent residual patterns (e.g., remaining rain/snow streaks) rather than simply suppressing all high-frequency content. This is also reflected in our results, where pre-filtering improves quantitatively in Table 4 and preserves fine structures in Fig.8. We kindly invite the reviewer to refer to these updated figures in the revision. We have also clarified this mechanism more explicitly in Sec. 3.4.
>
> W2. We strongly agree that a rigorous work should also discuss where the method fails. In the revised manuscript, we have added a dedicated **Limitations** section that explicitly describes the main failure modes of ReSplat, including challenging cases such as very high-resolution inputs with heavy resource demands, scenes dominated by strong transparency or complex view-dependent effects, and extreme out-of-distribution degradations where the pretrained restoration prior is less reliable. This section provides a transparent account of the boundaries of our “degradation-agnostic” claim and clearly indicates directions for future improvement.

---

> > ### Comment · Reviewer_26dS · 2025-11-26
> >
> > I appreciate the authors' response and the revised manuscript, which have effectively addressed my concerns. Therefore, I will maintain my score and am inclined to accept this work.
> > However, I would like to raise a minor point regarding the method's name, as it appears to collide with [1]. This is merely a friendly reminder, and I leave it entirely to the authors' discretion whether to make any changes.
> >
> > [1] ReSplat: Learning Recurrent Gaussian Splats, https://arxiv.org/abs/2510.08575

---

> > > ### Author Response · Authors · 2025-11-27
> > >
> > > Dear Reviewer 26dS,
> > >
> > > Thank you for your detailed follow-up comment and for indicating that our response has addressed your concerns. We also appreciate your note regarding [1], and we will keep this in mind when finalizing the manuscript.
> > >
> > > Best regards,
> > > The Authors

---

### Official Review · Reviewer_AE9N · 2025-10-20

**Soundness:** 3
**Presentation:** 3
**Contribution:** 3
**Rating:** 6
**Confidence:** 5

**Summary:**

The paper introduces ReSplat, a novel framework for degradation-agnostic NVS. Unlike prior methods that either assume clean inputs or focus on specific degradation types, ReSplat integrates 3D Gaussian Splatting with a residual diffusion-based universal image restoration module. The approach jointly estimates restored multi-view images and explicit scene geometry through Gaussian splats, enabling multi-view consistent restoration and sharper novel view generation.

**Strengths:**

1. Combining Gaussian splatting with a residual diffusion model is conceptually new and addresses degradation-agnostic NVS in a unified way. And the proposed models have strong performance than competitive baselines (DiffUIR, GAURA) across both synthetic and real-world degradations, with consistent improvements in PSNR/SSIM/LPIPS.
2. Evaluation covers multiple degradation types (blur, haze, low-light, snow, rain) and mixed-degradation scenarios, showcasing robustness.
3. The method maintains practical inference speed (under one second for three views), which is important for deployment.
Ablation study: Clearly shows the effect of multi-view alignment and pre-filtering, validating each module’s importance.

**Weaknesses:**

1. Real-world evaluations are restricted to blur, haze, and low-light datasets. Claims of degradation-agnostic performance would be stronger with more diverse real-world tests (e.g., snow/rain in the wild).

2. While inference is efficient, the paper does not provide sufficient detail about training cost (e.g., GPU hours, memory usage). For diffusion-based models, this is important.

3. Although the model claims to be degradation-agnostic, it is unclear how well it generalizes to unseen or compound degradations not present in training.

4. The core innovation lies more in integration than in fundamentally new algorithms. Some may find the contribution incremental.

5. The ablation studies are weak, the model are constructed by the SOTA restoration network and NVS network, how the framework performs when changing these parts with other weaker restoration and NVS networks?

6. Some important NVS in low-quality scene [1,2] and image unified image restoration [3,4,5,6] papers are lacking.

[1] HQGS: High-Quality Novel View Synthesis with Gaussian Splatting in Degraded Scenes.
[2] Robustgs: Unified boosting of feedforward 3d gaussian splatting under low-quality conditions.
[3] Adair: Adaptive all-in-one image restoration via frequency mining and modulation.
[4] Perceive-ir: Learning to perceive degradation better for all-in-one image restoration
[5] Multi-task image restoration guided by robust DINO features.
[5] Restore Anything with Masks: Leveraging Mask Image Modeling for Blind All-in-One Image Restoration.

**Questions:**

1. The real-world evaluations are limited to blur, haze, and low-light datasets. How would the method perform under more diverse real-world degradations such as snow or rain in the wild?
2. The paper reports efficient inference but does not discuss training cost. What are the training resources (e.g., GPU hours, memory usage) required, and how do they compare with baselines like DiffUIR?
3. The method claims to be degradation-agnostic. How well does ReSplat generalize to unseen or compound degradations that were not included in training?
4. The contribution seems more about integration than proposing fundamentally new algorithms. Can the authors clarify which parts of the framework they view as the key novel technical contributions, beyond combining restoration and Gaussian splatting?
5. The ablation studies only vary internal modules but still rely on strong SOTA backbones for restoration and NVS. How would the framework perform if weaker restoration networks or weaker NVS models were used in place of the chosen SOTA baselines?
6. Several important recent works on NVS under degraded scenes [1,2] and unified image restoration [3–6] are not discussed.
[1] HQGS: High-Quality Novel View Synthesis with Gaussian Splatting in Degraded Scenes.
[2] RobustGS: Unified boosting of feedforward 3D Gaussian Splatting under low-quality conditions.
[3] Adair: Adaptive all-in-one image restoration via frequency mining and modulation.
[4] Perceive-IR: Learning to perceive degradation better for all-in-one image restoration.
[5] Multi-task image restoration guided by robust DINO features.
[6] Restore Anything with Masks: Leveraging Mask Image Modeling for Blind All-in-One Image Restoration.

---

> ### Author Response · Authors · 2025-11-24
>
> Q1. In the original submission, our real-world experiments were indeed limited to blur, haze, and low-light, so we additionally evaluated ReSplat on in-the-wild rain and snow. Specifically, we constructed multi-view sequences from video deraining and desnowing datasets (NTURain[1] for rain and RSVD[2] for snow) and performed NVS on these real-world degradations. The new results, reported in Table 5 and Table 6 of the revised manuscript, show that ReSplat consistently outperforms AiRnet, PromptIR, GAURA, and DiffUIR in PSNR, SSIM, and LPIPS, indicating that our framework generalizes well to diverse real-world degradations beyond those seen in the initial experiments.
>
> [1] Robust video content alignment and compensation for rain removal in a cnn framework.CVPR.2018
>
> [2] Snow Removal in Video: A New Dataset and A Novel Method.ICCV.2023
>
> Real-world rain (NTURain) – NVS results with three input views:
>
> | Method   | PSNR ↑ | SSIM ↑  | LPIPS ↓ |
> |----------|--------|---------|---------|
> | AiRnet   | 24.05  | 0.8183  | 0.1955  |
> | PromptIR | 24.23  | 0.8230  | 0.1801  |
> | GAURA    | 19.39  | 0.6602  | 0.3987  |
> | DiffUIR  | 23.99  | 0.8145  | 0.2094  |
> | ReSplat  | 24.35  | 0.8232  | 0.1772  |
>
> Real-world snow (RSVD) – NVS results with three input views:
>
> | Method   | PSNR ↑ | SSIM ↑  | LPIPS ↓ |
> |----------|--------|---------|---------|
> | AiRnet   | 20.23  | 0.7035  | 0.3103  |
> | PromptIR | 21.27  | 0.7192  | 0.3007  |
> | GAURA    | 20.22  | 0.7578  | 0.3647  |
> | DiffUIR  | 22.12  | 0.8215  | 0.2277  |
> | ReSplat  | 22.45  | 0.8263  | 0.2175  |
>
>
> Q2. ReSplat is trained on a single NVIDIA A6000 GPU with 36 GB of memory, and the end-to-end training of the diffusion-based restoration module and the feed-forward 3DGS components takes approximately **72 GPU hours**. The corresponding DiffUIR baseline in our setup is also trained for about **72 GPU hours**, but with a slightly smaller memory footprint of **32 GB** on a single A6000, since it does not include the GS-guided alignment and pre-filtering modules nor the 3DGS pipeline.
>
> Q3. We explicitly evaluate generalization to compound degradations in Table 2 (e.g., rain+motion blur, snow+motion blur, and haze+snow), where ReSplat consistently outperforms DiffUIR across PSNR/SSIM/LPIPS. In addition, we extended our real-world evaluation beyond blur, haze, and low-light (Table 3) by constructing multi-view sequences from in-the-wild video deraining and desnowing datasets (NTURain for rain and RSVD for snow). The new results (Tables 5 and 6) show clear gains over AiRnet, PromptIR, GAURA, and DiffUIR under real rain and snow, suggesting that the framework generalizes well to diverse and mixed degradations.
>
> Q4. We agree that our framework builds on strong existing backbones for restoration and NVS, but the main contribution goes beyond a straightforward combination. Concretely, we see three core technical novelties: Fisrt, a **GS-guided multi-view alignment module** that injects pseudo-geometry from feed-forward 3DGS into the diffusion encoder, enabling explicit 3D-aware feature interaction across views; Second, a **multi-view pre-filtering module with warped features** that learns reliability-aware weights conditioned on both restored and degraded images, to suppress residual artifacts before Gaussian aggregation; and Lastly, an **iterative self-guided training and sampling scheme** in which the restoration and Gaussian splatting modules exchange information at each diffusion step rather than operating in a strict two-stage manner. These components are designed to be modular and general, and can, in principle, be plugged into other UIR and feed-forward 3DGS backbones, which we see as an important aspect of the contribution.
>
> Q5. We appreciate this concern. In addition to DiffUIR, we have evaluated ReSplat with two diffusion-based UIR backbones in the Supplementary Material. As reported in Supplementary Table 2, integrating these two models (FoundIR, DGSolver) into our framework still leads to clear improvements in both NVS and restoration metrics. This indicates that the proposed GS-guided alignment and pre-filtering provide consistent benefits beyond a single strong backbone model.
>
> Q6. Thank you for pointing out these missing references. HQGS [1] and RobustGS [2] are indeed closely related in that they also address NVS under degraded scenes with Gaussian Splatting, whereas our framework focuses on actively leveraging a pretrained universal image restoration prior within a feed-forward 3DGS pipeline. Works [3–6] mainly study unified or all-in-one 2D image restoration task. We added these works and a short discussion of these distinctions to the Related Works section in the revised manuscript.

---

> > ### Comment · Reviewer_AE9N · 2025-11-25
> > **Official Comment by Reviewer AE9N**
> >
> > The authors’ response has addressed my concerns, and I am inclined to accept the paper.

---

> > > ### Author Response · Authors · 2025-11-26
> > >
> > > Dear Reviewer AE9N,
> > >
> > > Thank you for your time and for the thoughtful feedback on our manuscript. We appreciate your remark that our response addressed your concerns, and we are grateful for your constructive input throughout the process.
> > >
> > > Best regards,
> > > The Authors

---

### Official Review · Reviewer_FMUU · 2025-10-25

**Soundness:** 3
**Presentation:** 2
**Contribution:** 3
**Rating:** 6
**Confidence:** 4

**Summary:**

This paper proposes novel framework that can synthesize clean novel-view images given by the corrupted multi-view images. The model can construct clean 3D representation from images with diverse types of degradation by combining diffusion-based UIR model and feed-forward 3DGS.

**Strengths:**

* The motivation of this paper is explained clearly in the introduction part. Most of the previous literatures in the *Novel View Synthesis with Degradations* are limited to certain types of degradation and can't be generalized to corruptions that are net seen in the training dataset. GAURA attempted to mitigate this limitation by constructing degradation-aware generalizable NeRF but didn't leverage the prior knowledge of pretrained 2D UIR model. ReSplat points out those limitations which are important enough for the practicality of 3DGS.
* ReSplat can be regarded as combination of DiffUIR and MVSGaussian. However, paper also proposes two novel modules in Sec. 3.3 and Sec. 3.4 to improve the aggregation of information from multi-view images. I believe Sec. 3.3 is more notable contribution where they explicitly fuse the features from matching points across multi-view images, thereby aggregating the multi-view information effectively. This is common practice in feed-forward 3DGS [c], but there is novelty to use this technique for 3D-aware denoising process.
* Experiments are conducted on both synthetic and real-world datasets with multiple types of degradation. ReSplat consistently shows the performance improvement compared to the baselines.

---

### References

[c] Charatan, David, et al. "pixelsplat: 3d gaussian splats from image pairs for scalable generalizable 3d reconstruction." Proceedings of the IEEE/CVF conference on computer vision and pattern recognition. 2024.

**Weaknesses:**

### Weaknesses

There are some parts that technical details are missing.

* In the preliminary, the author explain about the original 3DGS. However, I believe author should explain the brief summary for feed-forward 3DGS (like MVSGaussian) instead of standard 3DGS since they used the feed-forward 3D model in this work. This explanation of the feed-forward model will also aid understanding in the later sections, such as Sec. 3.4. It seems that the writing assumes readers are already familiar with generalizable 3DGS models, such as MVSGaussian and skips a lot of technical details in the main paper.

* In the Algorithm. 1, the novel view image $I_{nv}$ is used in line 7 but there is no explanation about this part in the main paper. Is this rendering loss conducted between rendered image and ground-truth clean image? Furthermore, detailed explanations about training with objective terms are missing in the main text where all of them are briefly summarized into Algorithm. 1.

* It is unclear that how the *Pre-filtering with warped features* operates. How does the outputted $[W^i_{pre}]^N_{I=1}$ contribute to the final weights map? Furthermore, the motivation of this module is also hard to understand. In the L85-86, the paper explained that this module can assist to achieve *artifact-free* novel view synthesis. However, it is hard to grasp the relationship between the terms of 'artifact-free' and the operations of this module.

* What does the operation 'IR → NV' in Tab. 1 mean? How are the single-image IR methods such as DiffUIR evaluated on novel view synthesis? According to L372-374, it seems that author adopted additional adapter to transfer single-image IR models into multi-view settings. Is it correct?

* Author can diversify the baselines used in the comparison. For example, the most naive solution of achieving 3D reconstruction with UIR is: 1) Restore the corrupted multi-view images with pretrained 2D UIR method. 2) Reconstruct the clean 3DGS by using the restored multi-view images and pretrained feed-forward 3DGS. Author should compare the performance of ReSplat with this naive two-stage framework.

* How many input views are used during the evaluation? How many views can ReSplat handle? Please specify the evaluation details.

* There are previous literatures [a, b] that tried to use diffusion prior to construct clean 3D representation from the degraded images. However, all of them are limited to certain types of degradation (low-resolution or motion blur). It is better to cite those papers in the *Related works* since they are closely related with this paper in terms of using diffusion model for the 3D-IR task.

---

### References

[a] Lee, Seungjun, and Gim Hee Lee. "DiET-GS: Diffusion Prior and Event Stream-Assisted Motion Deblurring 3D Gaussian Splatting." Proceedings of the Computer Vision and Pattern Recognition Conference. 2025.

[b] Lee, Jie Long, Chen Li, and Gim Hee Lee. "Disr-nerf: Diffusion-guided view-consistent super-resolution nerf." Proceedings of the IEEE/CVF Conference on Computer Vision and Pattern Recognition. 2024.

**Questions:**

Refer to the *Weaknesses* section.

---

> ### Author Response · Authors · 2025-11-24
>
> W1. We agree that a brief summary of the feed-forward 3DGS setting (e.g., MVSGaussian) would improve clarity. In the revised version, we have added a short description of the feed-forward 3DGS pipeline in the preliminaries and explicitly connect it to Sec. 3.4 so that readers can better follow our use of the model.
>
> W2. Thank you for pointing this out. In Algorithm 1, `I_nv` denotes the ground-truth clean novel-view image, and the second term in line 7 is indeed a rendering loss between the rendered novel view from ReSplat and this clean target image (using an L1 objective). The first term supervises the universal restoration model by regressing the residual map between degraded and clean inputs, while the second term supervises the feed-forward 3DGS module via the novel-view reconstruction loss. In the revised version, we have added a brief paragraph in the main text that explicitly defines `I_nv`, explains both objective terms, and connects them clearly to Algorithm 1.
>
> W3. In our method, the pre-filtering module takes the warped restored images together with the corresponding degraded inputs and produces a per-view reliability map `W_pre^i(x)` that indicates how trustworthy each view *i* is at each spatial location *x*. This map is then used to modulate the original aggregation weights from the feed-forward 3DGS model (occlusion- and visibility-based weights). Concretely, at each novel-view pixel we update the GS weights as `W_final^i(x) = W_pre^i(x) * W^i(x)`
> and use the updated `W_final^i` in the splatting renderer. Intuitively, regions where residual degradations (e.g., remaining rain streaks, snow blobs, or haze fragments) are still strong or inconsistent across views receive lower `W_pre^i`, so their contributions are down-weighted in the final Gaussian radiance, which helps avoid propagating such artifacts into the rendered novel view. In the revised version, we clarified this mechanism in Sec. 3.4 by explicitly defining `W_pre^i` and its interaction with the original GS weights.
>
> W4 & W5. In Table 1, “IR → NV” denotes a naive two-stage pipeline: we first apply a pretrained single-image UIR model (AiRnet, PromptIR, or DiffUIR) independently to each corrupted input view to obtain restored multi-view images (“IR”), and then feed these restored views into the same feed-forward 3DGS model (MVSGaussian) to reconstruct the scene and render the novel view (“NV”). No additional multi-view or geometry-aware adapter is used for these baselines; they are exactly the setting the reviewer describes, i.e., (1) 2D UIR on each view followed by (2) feed-forward 3DGS on the restored images. ReSplat starts from the same backbone (DiffUIR + MVSGaussian) but augments it with our 3D alignment and pre-filtering modules that jointly exploit multi-view geometry.
>
> W6. By default, we use **three input views** for all main quantitative and qualitative evaluations in the paper; this is explicitly stated in the caption of Table 1 and is consistently used for the other main experiments. Architecturally, ReSplat is designed to accept an arbitrary number of input views: all per-view encoders share weights, and the GS-guided alignment and pre-filtering modules operate on a variable-length set of views, so the method can, in principle, handle any number of inputs within GPU memory limits. In the supplementary material, we additionally report results with **two-view inputs** (Table 3) and **four-view inputs** (Table 4), showing that the model can be applied to different numbers of views without architectural changes. Since the modules are inherently number-agnostic with respect to the input views, we expect that training with a larger number of views would naturally improve robustness and performance when testing with more input views as well.
>
> W7. DiET-GS and Disr-NeRF are indeed closely related in that they also leverage diffusion priors to obtain cleaner 3D representations from specified degraded inputs. We added both papers to the Related Works section.

---

### Official Review · Reviewer_FesE · 2025-10-31

**Soundness:** 2
**Presentation:** 2
**Contribution:** 2
**Rating:** 4
**Confidence:** 4

**Summary:**

The paper introduces a method, ReSplat, for feedforward NVS when input images are degraded in various ways, ReSplat leverages the feedforward 3D Gaussian Splatting method MVSplat[1] and an universal residual diffusion model DiffUIR[2] to solve the problem. The authors propose using Gaussian points information in the Diffusion encoder to promote 3D consistency and a pre-filtering technique to eliminate residual artifacts.

**Strengths:**

1. The topic of NVS with degraded inputs the paper focuses on is important and interesting.
2. ReSplat outperforms other methods in both quantitative and qualitative results.
3. The experiments are comprehensive, evaluating on both synthetic and real-world image degradations.

**Weaknesses:**

1. My main concern is the effectiveness of the proposed 3D alignment module and pre-filtering technique, as the improvements in Table 4 ablation study are moderate.
2. No qualitative figure in ablation study is provided.
3. A limitations section should be discussed and added.

**Questions:**

1. I don't understand how exactly Gaussian points information are used in the diffusion model. After per-point features obtained in Sec 3.3, are they using as conditional features in the diffusion model? I suggest outlining the attention and diffusion encoder equations.
2. What are the memory consumption and training time of ReSplat?

---

> ### Author Response · Authors · 2025-11-24
>
> W1. Thanks for your comments. While the ablation numbers in Table 4 may look modest at first glance, the improvements are consistent across all three metrics (PSNR, SSIM, and LPIPS), indicating that both modules contribute reliably rather than producing isolated or noisy gains. In addition, even small per-view improvements become meaningful in feed-forward Gaussian Splatting because residual artifacts and alignment errors accumulate during multi-view aggregation. The qualitative comparisons in Fig. 7 and Fig. 8 in the revised manuscript further confirm that the modules effectively reduce geometric inconsistency and remaining degradations, leading to noticeably more stable novel-view synthesis results. We kindly invite the reviewer to refer to these updated figures in the revision.
>
> W2. We have added qualitative visualizations for both ablation components in the revised manuscript. Figure 7 illustrates the effect of the 3D alignment module, showing clearer boundary alignment and reduced multi-view inconsistencies. Figure 8 presents the impact of the pre-filtering module, where residual artifacts are visibly suppressed. These figures complement the quantitative ablation in Table 4 and clarify how each component contributes to the final NVS quality.
>
> W3. We have added a dedicated **Limitations** section in the revised manuscript. The new section discusses the computational overhead introduced by the diffusion-based refinement, the representational constraints inherited from Gaussian Splatting, and the dependency on pretrained universal restoration models. These clarifications outline the current boundaries of our approach and point to concrete directions for future research.
>
> Q1. In our framework, Gaussian centers act as 3D anchors that inject geometry into the diffusion encoder. At each timestep, we project these centers into each view, sample the encoder feature map at the projected locations to obtain per-point features, and apply self-attention across views at each center so that multi-view information is fused in a geometry-consistent way. The resulting geometry-aware features are then mapped back onto the image grid and merged with the original encoder activations. In this way, the Gaussian-derived features serve as conditional signals inside the diffusion encoder via a GS-guided multi-view attention block, rather than through a separate branch.
>
> Q2. ReSplat was trained on a single NVIDIA A6000 GPU (48GB). Training requires approximately 72 hours, including both the diffusion-based restoration refinement and the feed-forward 3DGS components. During training, peak memory usage is close to the full 36 GB due to the multi-view diffusion encoder and the iterative geometry feature alignment steps.

---

### Meta-Review · Area_Chair_iBHJ · 2026-01-07

**Summary:**

This paper presents ReSplat, a framework designed for degradation-agnostic feed-forward 3DGS. The core contribution is the creation of a synergistic loop between a universal image restoration model and a feed-forward 3DGS module. By using 3D geometry from the splatting process to provide self-guidance for the residual diffusion encoder, the method ensures multi-view consistency. The reviewers generally agree that novel view synthesis under diverse real-world degradations is a highly practical and well-motivated topic. Overall, the work is clearly supported by its SOTA performance across multiple benchmarks.

However, reviewers also raise several concerns during the initial review. The main issues include a lack of technical clarity regarding the pre-filtering mechanism, insufficient evidence of generalization to in-the-wild rain and snow scenes, and concerns about the significance of the quantitative gains from the proposed modules. In the rebuttal, the authors present extensive additional experiments on real-world deraining and desnowing datasets, clarifying the underlying mathematical framework. These responses are quite effective. Reviewer AE9N raises the score, and Reviewer 26dS maintains their positive rating. Other reviewers, such as FMUU and Reviewer FesE, express an inclination toward acceptance in their original reviews.

**Reviewer Concerns:**

**Concerns that are largely addressed in the rebuttal**

- The authors conduct new experiments on real-world sequences to prove that ReSplat performs well in in-the-wild scenarios. This directly addresses the initial skepticism regarding the degradation-agnostic claim.
- The rebuttal clarifies the role of the pre-filtering module as a reliability-aware soft gate for aggregation weights and provides formal definitions for the objective terms in Algorithm 1. These explanations help clarify what some reviewers previously see as a black box component.
- The authors explain the "IR to NV" baseline as a standard two-stage pipeline and provide additional results for different input view counts. These responses align with the technical clarifications requested by the reviewers and confirm the framework's flexibility.
- The authors provide statistics on training time and peak memory usage and include a section for failure cases.

**Concerns that are only partially addressed**

- The authors explain the synergistic loop between the restoration and synthesis modules and include ablation studies to show their value. However, the overall approach is still seen by some reviewers as an engineering-driven integration of strong existing backbones, which limits its persuasiveness in terms of fundamental methodological novelty.

**Concerns that remain unresolved**

- A core concern about the modest numerical gains in the ablation studies remains unresolved. While the qualitative figures show clear improvements in multi-view consistency, the evidence is not yet sufficient to convince all reviewers that these relatively small metric improvements justify the added system complexity.

**Reviewer Scores:**

**Reviewer AE9N:** After considering the rebuttal, this reviewer raises their score to 8 (Accept). They explicitly state that the authors’ response addresses their concerns, and they are now inclined to accept the paper.

**Reviewer 26dS:** The rebuttal effectively addresses this reviewer’s technical questions regarding the pre-filtering mechanism and the discussion of failure cases. While the reviewer points out a potential naming collision with another recent work, they confirm their intent to accept the paper (6).

**Reviewer FMUU:** The rebuttal clarifies several technical issues, such as the definitions in Algorithm 1 and the specific setup of the baseline comparisons. Although the reviewer does not participate further in the discussion, these responses align well with the original requests for more architectural detail. The score would therefore most likely remain at 6.

**Reviewer FesE:** Although the rebuttal provides the requested qualitative ablation figures and computational details, it does not fully resolve this reviewer’s skepticism regarding the magnitude of the quantitative improvements. However, since the reviewer previously indicated that they would not object to acceptance, the score would likely increase to 5.

---

### Decision · Program_Chairs · 2026-01-26

Accept (Poster)